# Infralimbic parvalbumin neural activity facilitates cued threat avoidance

Yi-Yun Ho[1,2], Qiuwei Yang[1], Priyanka Boddu[1], David A Bulkin[1,2], Melissa R Warden[1,2,3,4]*

[1]Department of Neurobiology and Behavior, Cornell University, Ithaca, United States; [2]Cornell Neurotech, Cornell University, Ithaca, United States; [3]Department of Translational Neurosciences, University of Arizona College of Medicine, Phoenix, United States; [4]Graduate Interdisciplinary Program in Neuroscience, University of Arizona, Tucson, United States

## eLife assessment

This **important** study extends our understanding of how the medial prefrontal cortex regulates flexible action during adversity. The data provide **compelling** evidence of a role for prefrontal PV neuron activity in active avoidance. This builds on the general idea that these neurons play a role in flexible behavior and demonstrates this in the context of freezing/avoidance conflict. The overall findings contribute to our understanding of mechanisms that support aversively motivated instrumental learning and may provide insight into both stress vulnerability and resilience processes. This work will be of interest to those interested in learning, aversive motivation, interneuron and/or prefrontal cortex function, or conditions relates to these processes and mechanisms.

*For correspondence:
mrwarden@gmail.com

**Abstract** The infralimbic cortex (IL) is essential for flexible behavioral responses to threatening environmental events. Reactive behaviors such as freezing or flight are adaptive in some contexts, but in others a strategic avoidance behavior may be more advantageous. IL has been implicated in avoidance, but the contribution of distinct IL neural subtypes with differing molecular identities and wiring patterns is poorly understood. Here, we study IL parvalbumin (PV) interneurons in mice as they engage in active avoidance behavior, a behavior in which mice must suppress freezing in order to move to safety. We find that activity in inhibitory PV neurons increases during movement to avoid the shock in this behavioral paradigm, and that PV activity during movement emerges after mice have experienced a single shock, prior to learning avoidance. PV neural activity does not change during movement toward cued rewards or during general locomotion in the open field, behavioral paradigms where freezing does not need to be suppressed to enable movement. Optogenetic suppression of PV neurons increases the duration of freezing and delays the onset of avoidance behavior, but does not affect movement toward rewards or general locomotion. These data provide evidence that IL PV neurons support strategic avoidance behavior by suppressing freezing.

## Introduction

The prefrontal cortex is essential for flexible behavior (*Duncan, 1986*; *Miller and Cohen, 2001*). Hallmarks of prefrontal damage in humans and animals include stimulus-bound and context-inappropriate behaviors, excessive reactivity, and impulsivity (*Harlow, 1868*; *Bianchi and Macdonald, 1922*; *Lhermitte, 1983*). The ventromedial part of the prefrontal cortex, the infralimbic cortex (IL), is important for supporting strategic behavior in the face of environmental threat (*Murphy et al., 2005*; *Hardung et al., 2017*). IL plays a critical role in fear extinction (*Milad and Quirk, 2002*; *Do-Monte et al., 2015*),

discrimination between safety and fear (*Sangha et al., 2014*; *Sangha et al., 2020*), and active avoidance (*Moscarello and LeDoux, 2013*; *Halladay and Blair, 2017*).

During active avoidance, mice first freeze in response to shock-predicting tones, as they do in fear conditioning, but gradually learn that they can avoid the shock by crossing the chamber when a tone plays. This behavior requires both the suppression of cued freezing and movement toward a safe zone (*Mowrer and Lamoreaux, 1946*; *Koolhaas et al., 1999*; *Moscarello and LeDoux, 2013*; *Moscarello and LeDoux, 2014*; *Krypotos et al., 2015*; *LeDoux et al., 2017*). Active avoidance is similar to fear extinction in that cue-elicited freezing behavior mediated by the amygdala is suppressed during both behaviors (*Phillips and LeDoux, 1992*; *Kim et al., 1993*; *Herry et al., 2010*; *Milad and Quirk, 2012*; *Moscarello and LeDoux, 2013*; *LeDoux et al., 2017*). IL neural activity is higher in rats that successfully extinguish freezing to conditioned stimuli, and IL projects directly and indirectly to the central amygdala and is thought to suppress central amygdala outputs that mediate freezing (*Milad and Quirk, 2002*; *Vertes, 2004*; *LeDoux et al., 2017*).

IL parvalbumin (PV) neurons synapse onto and inhibit local pyramidal neurons. Although we might expect that activation of IL PV neurons would inhibit avoidance behavior by inhibiting IL long-range projection neurons and disinhibiting central amygdala outputs that facilitate freezing, the relationship between PV and pyramidal neuron firing in cortex is complex. Cortical PV neurons have been reported to activate simultaneously with local pyramidal neurons (*Merchant et al., 2008*; *Okun and Lampl, 2008*; *Isomura et al., 2009*; *Pinto and Dan, 2015*; *Estebanez et al., 2017*; *Nashef et al., 2022*; *Giordano et al., 2023*), and it has been suggested that PV neurons may help to shape, rather than gate, the firing of local pyramidal neurons (*Isomura et al., 2009*; *Merchant et al., 2012*).

The question thus arises whether the activation of IL PV neurons suppresses or facilitates active avoidance behavior. Using fiber photometry, we show that IL PV neuron activity increases specifically when mice suppress cue-elicited freezing and move to avoid a future shock, but does not increase when animals move to obtain a cued reward or move in a neutral context. Further, we show that movement-related IL PV neural activity precedes avoidance learning, and emerges after mice have experienced a single shock in an environment, a finding that links PV neural activity specifically with the suppression of freezing to enable movement. Finally, we show that optogenetic suppression of IL PV neural activity prolongs freezing and delays avoidance but does not affect movement toward cued rewards or general locomotion. These results reveal that IL PV neurons play an essential and counter-intuitive role in supporting flexible behavior in the face of threat, and suggest a role for PV neurons in shaping IL function that goes beyond the suppression of local neural activity.

## Results

### IL PV neurons signal active avoidance

We first asked how IL PV neurons respond during active avoidance behavior. To target this population for fiber photometry, we selectively expressed a genetically encoded calcium indicator in IL PV neurons by injecting AAV-CAG-Flex-GCaMP6f into IL in PV-Cre mice (*Hippenmeyer et al., 2005*; *Chen et al., 2013*). In control mice, we expressed GFP in IL PV neurons by injecting AAV-CAG-Flex-GFP. We implanted an optical fiber over IL to monitor calcium-dependent fluorescence, and recorded IL PV population activity via fiber photometry (*Figure 1A and B*; *Cui et al., 2013*; *Gunaydin et al., 2014*).

We used a two-way signaled active avoidance paradigm (*Figure 1C*). When an auditory cue (constant tone at 12 kHz or 8 kHz) was played, mice were required to cross the midline of the behavioral testing chamber within 5 s of tone onset to avoid an impending foot shock. If mice crossed the chamber during this 5 s period, no foot shock was delivered and the trial was scored as a successful avoidance. If not, a 2 s foot shock was delivered. Mice could terminate the foot shock early by crossing the chamber, which was scored as an escape. The tone terminated at either successful avoidance or shock offset. Prior to avoidance training, mice received two tone-shock pairings to learn the association between the auditory cue and the foot shock.

As IL inactivation impairs avoidance (*Moscarello and LeDoux, 2013*; *Halladay and Blair, 2017*), we predicted that the activity of inhibitory IL PV neurons would be low during successful avoidance trials. Contrary to expectations, we found that IL PV neural activity rose upon the initiation of avoidance movements and peaked at chamber crossing (*Figure 1D–G and G*: N=8 mice, p=0.0077, paired t-test; *Figure 1—figure supplement 1A–F,B*: N=8 mice, p=0.0013, paired t-test, **F**: N=8 mice, p=0.4682,

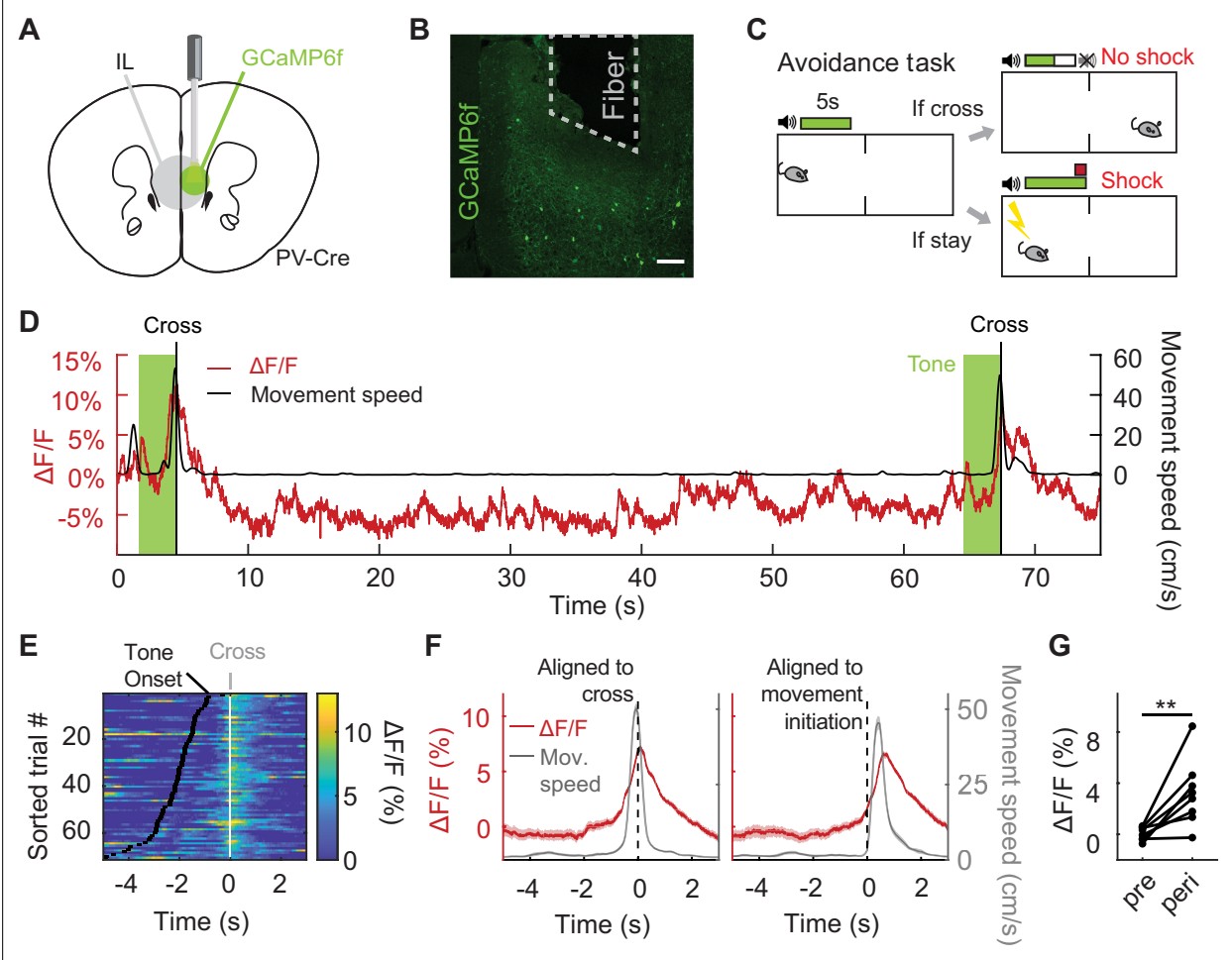

**Figure 1.** IL PV neurons signal active avoidance. (**A**) Fiber photometry schematic. (**B**) GCaMP6f expression in IL PV neurons. Scale bar, 100 μm. (**C**) Active avoidance task schematic. The green box indicates presence of the tone, which lasts till animal crosses. (**D**) Example IL PV ΔF/F (red) and speed (black) during two successful avoidance trials. Vertical line indicates chamber crossing. Tone, light green. (**E**) IL PV ΔF/F during successful avoidance trials, data aligned to chamber crossing. White ticks: chamber crossing. Black ticks: tone onset. Same example mouse as D. (**F**) Example average IL PV ΔF/F (red) and speed (grey), aligned to chamber crossing (left) and movement initiation (right). Same example mouse as D. (**G**) Average IL PV ΔF/F before chamber crossing (pre, 4–2 s before cross) and during chamber crossing (peri, 1 s before to 1 s after crossing). **p<0.01, paired t-test. Shaded regions indicate SEM.

The online version of this article includes the following figure supplement(s) for figure 1:

**Figure supplement 1.** IL PV neurons respond specifically to avoidance movements.

paired t-test). The trial-by-trial variability in the amplitude of IL PV neural activity was not correlated with variability in the speed or latency of the avoidance movement (*Figure 1—figure supplement 1G–L, H*: N=8 mice, p=0.3541, one-sample t-test, **K**: N=8 mice, p=0.4132, one-sample t-test). IL PV neural activity during the avoidance movement was not attenuated by learning or repeated reinforcement (*Figure 1—figure supplement 1M and N*, N=8 mice, p=0.8886, one-way ANOVA).

## IL PV neural activity reflects the avoidance movement, not the predictive tone

During successful avoidance trials, two events happen simultaneously: the mouse crosses the chamber, and a shock-predicting tone is terminated. To determine whether PV neural activity better reflects the avoidance movement or the termination of the predictive auditory sensory cue, we designed a version of the avoidance task with additional trial types to uncouple these events. Regular trials (80%) were interleaved with trials with shortened (10%) or lengthened (10%) tones (*Figure 2A*). In short-tone trials the tone lasted for only 1.5 s, and in long-tone trials the tone was not terminated until

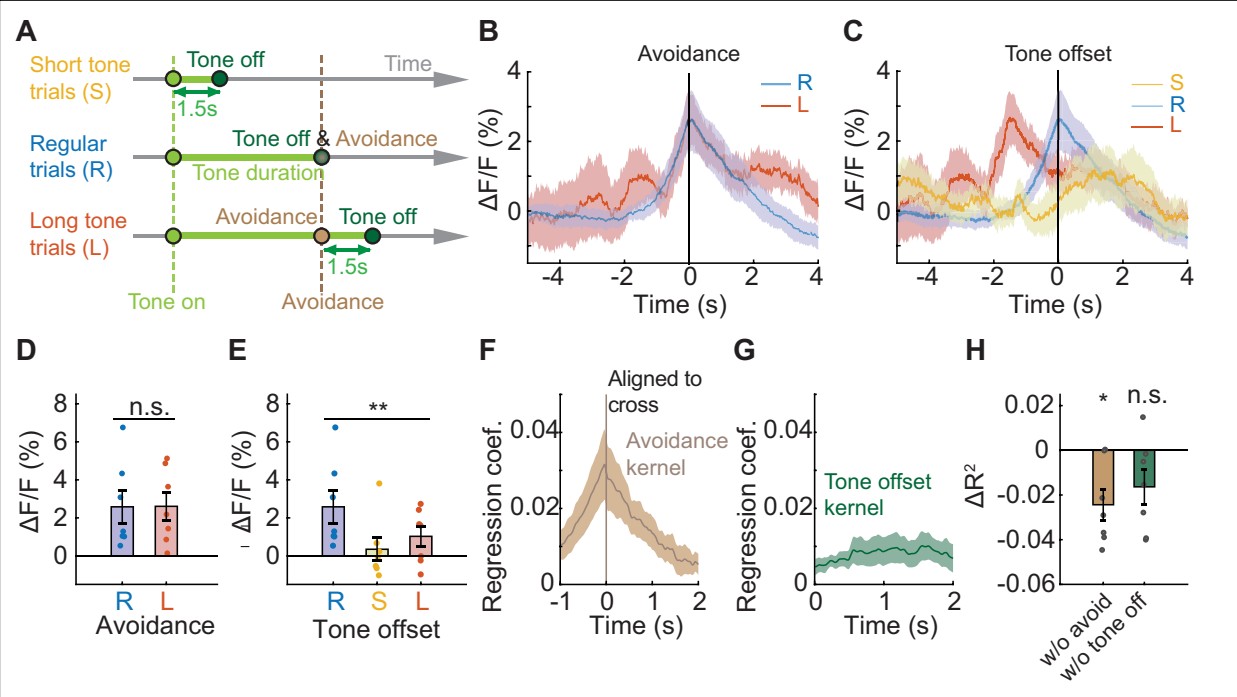

**Figure 2.** IL PV neural activity reflects the avoidance movement, not the predictive tone. (**A**) Schematic of a modified version of avoidance task including 10% short tone trials (S), where the tone is always 1.5 s; 10% long tone trials (L), where the tone lasts until 1.5 s after successful avoidance; and 80% regular trials (R), where the tone terminates upon successful avoidance. (**B–C**) Average IL PV ΔF/F aligned to (**B**) chamber crossing and (**C**) tone offset. (**D**) Comparison between average IL PV calcium activity 0–0.1 s after avoidance chamber crossings during regular trials and long tone trials (ns = non-significant, paired t-test). (**E**) Comparison between average IL PV calcium activity 0–0.1 s after tone offsets during regular trials, short tone trials, and long tone trials (**p<0.01, one-way repeated ANOVA). (**F–G**) Average avoidance kernel (**F**) and average tone offset kernel (**G**) across all animals calculated by a linear model. (**H**) Loss of predictive power (ΔR²) in a reduced model with shuffled avoidance or tone offset time points (ns = non-significant, *p<0.05, one-sample t test). Error bars and shaded regions indicate SEM.

1.5 s after successful avoidance (*Figure 2A*). IL PV neural activity at chamber crossing did not differ between regular and long-tone trials (*Figure 2B and D*, N=7 mice, p=0.9546, paired t-test). Usually, the chamber was not crossed in short-tone trials, so short-tone trials were not included in this analysis. PV neural activity at tone offset was significantly different among short-tone, regular, and long-tone trials (*Figure 2C and E*, N=7 mice, p=0.005, one-way repeated ANOVA. Multiple comparison, R and S, p=0.0045, R and L, p=0.0398, S and L, p=0.4620.). These results indicate that PV neural activity primarily reflects the avoidance movement and not tone offset.

We extracted characteristic neural responses to each event without interference from other events close in time. We constructed a linear regression model to extract the isolated neural responses to chamber crossing and tone termination (*Parker et al., 2016*; *Musall et al., 2019*), and all major events were included in the model. We assumed (1) that neural responses would be similar for the same events and dissimilar for different events, and (2) neural responses to events can be summed up linearly to form the recorded signals. With these assumptions and through linear regression, isolated neural responses to each event were extracted. The model-extracted isolated neural response to tone offset was flat, while the isolated avoidance signal peaked at chamber crossing (*Figure 2F and G*). To further demonstrate that IL PV activity can be better accounted for by the action to avoid the shock than tone termination, we compared the explanatory power (R 2, coefficient of determination) of the reduced model with shuffled time points to the full model. Shuffling the avoidance time points significantly reduced the explanatory power of the model (ΔR 2 different from zero, N=7 mice, p=0.0125, one-sample t-test), while shuffling the tone offset time points had no effect (*Figure 2H*, N=7 mice, p=0.0802, one-sample t-test). Thus, IL PV neural activity at chamber crossing reflects the avoidance behavior and not the cessation of the predictive auditory cue.

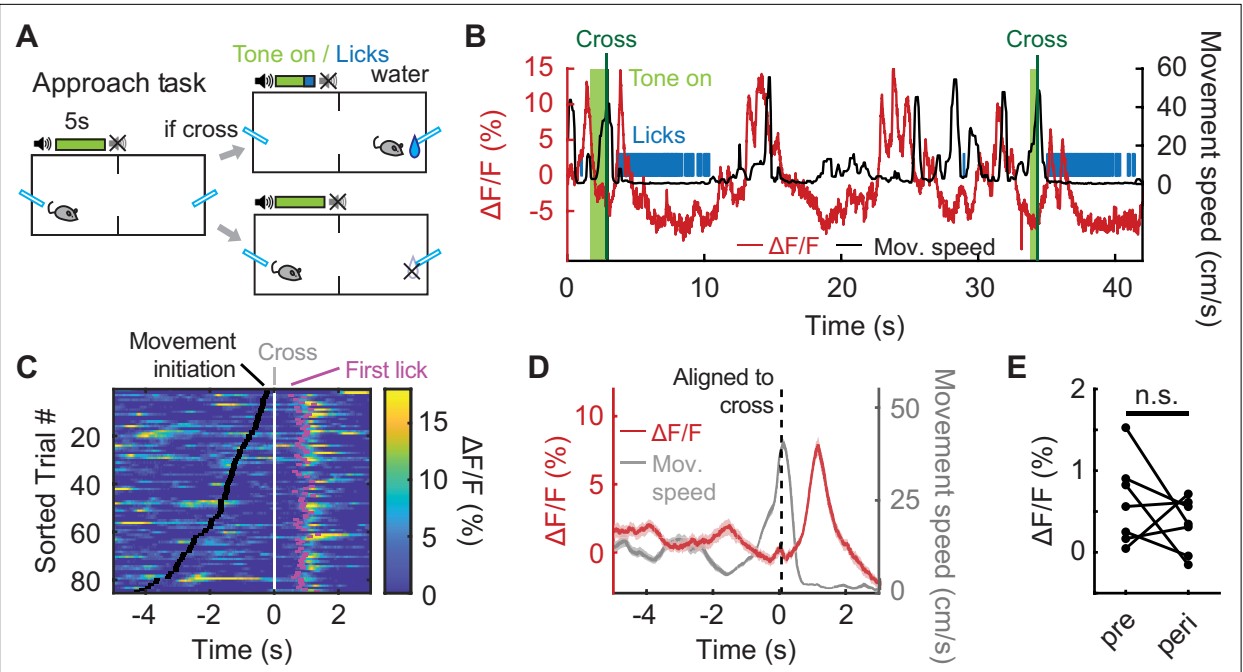

**Figure 3.** IL PV neural activity does not reflect movement to obtain rewards. (**A**) Reward approach task schematic. (**B**) Example IL PV ΔF/F (red) and speed (black) during two successful approach trials. Vertical line indicates chamber crossing. Tone, light green. (**C**) IL PV ΔF/F during successful approach trials, data aligned to chamber crossing. Black ticks: tone onset; white ticks: chamber crossing; magenta ticks: first licks. Same example mouse as B. (**D**) Example average IL PV ΔF/F (red) and speed (grey), aligned to chamber crossing (left). Same example mouse as B. (**E**) Average IL PV ΔF/F before chamber crossing (pre, 4–2 s before cross) and during chamber crossing (peri, 1 s before to 1 s after crossing). ns = non-significant, paired t-test. Shaded regions indicate SEM.

The online version of this article includes the following figure supplement(s) for figure 3:

**Figure supplement 1.** IL PV neural activity does not reflect reward receipt.

**Figure supplement 2.** IL PV neural activity does not reflect movement in the OFT.

## IL PV neural activity does not reflect movement to obtain rewards

IL PV neural activity rises during movement to avoid a predicted future shock (*Figure 1E and F*), but it is unclear whether this neural activity specifically reflects avoidance, or if IL PV neural activity would be elevated during any movement. If IL PV neural activity reflects locomotor activity, we would expect to see elevated neural activity during movement to obtain rewards. To test this idea, we recorded IL PV neuronal activity during a reward approach task, which mirrored the avoidance design in temporal structure, but shock omission on successful chamber crossing was replaced with reward delivery (*Figure 3A*). In this task, chamber-crossing movements to obtain a water reward were not associated with elevated IL PV neural activity (*Figure 3B-E*; E, N=7 mice, p=0.3003, paired t-test; *Figure 3— figure supplement 1A-C*, C: N=6 mice, p=0.0382, paired t-test; see *Figure 1C-G* for comparison).

We observed an elevation of IL PV neuronal activity after chamber crossing when the animals started to consume the water reward (*Figure 3C*, magenta dots, and *Figure 3D*). To test whether this transient increase in IL PV neuron activity after chamber crossing reflects water consumption or the preceding approach behavior, we introduced randomized water-omission trials in 10% of the trials (*Figure 3—figure supplement 1D–E*). We found that the PV neuron signal rose both when water was received and when the water reward was omitted, and increases in IL PV activity preceded the first lick to consume water reward delivered after a successful crossing (*Figure 3—figure supplement 1E*). We speculate that, rather than a reward signal related to water consumption, this signal may reflect the suppression of the habitual chamber-crossing movement in the approach task when the mouse nears the lick port.

To further investigate whether elevated IL PV neuronal activity reflected avoidance or locomotion, we recorded IL PV activity in an open field test (OFT) where animals were allowed to run freely without any defined task structure (*Figure 3—figure supplement 2A and B*). When we aligned IL PV activity

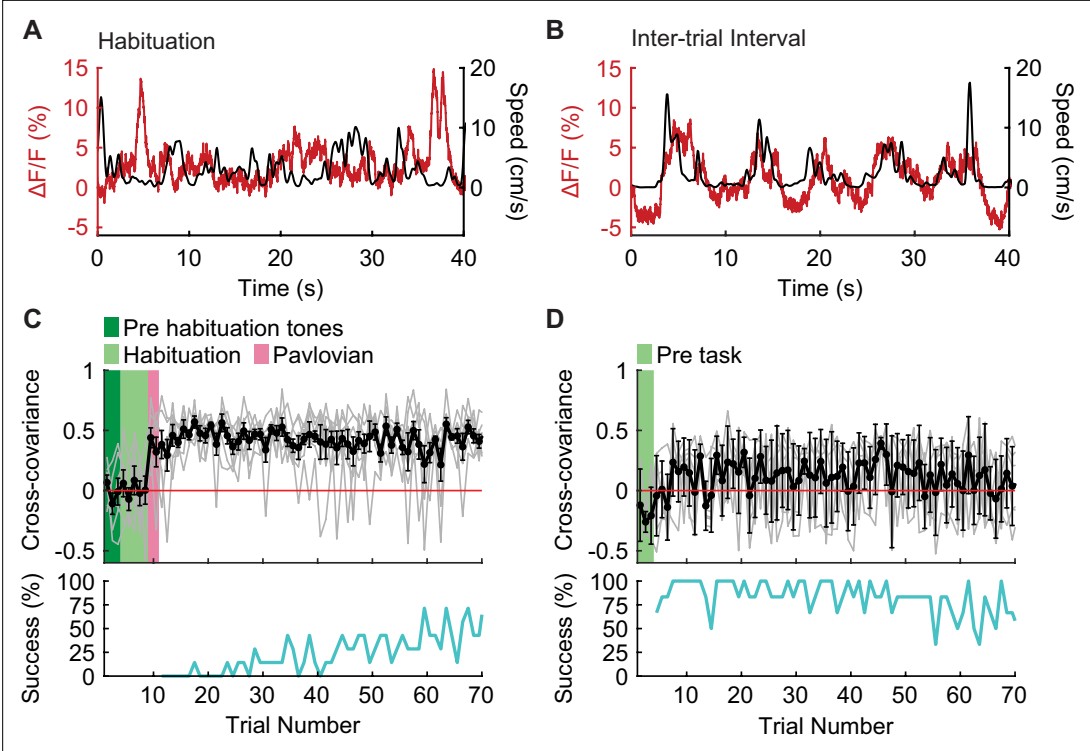

**Figure 4.** IL PV neural activity becomes correlated to movement after shock. (**A–B**) Example IL PV ΔF/F (red) and speed (black) during (**A**) habituation prior to the first-ever shock exposure, and (**B**) the intertrial interval after avoidance training. (**C–D**) Evolution of cross-covariance between IL PV activity and speed over trials (upper panel, black) and corresponding average avoidance success rate across animals (bottom panel, cyan) (**C**) on the first day of avoidance training (N=7 mice) and (**D**) during approach in well-trained mice (N=6 mice, green shading marks the time in chamber before the task [pre-task]). Error bars indicate SEM.

The online version of this article includes the following figure supplement(s) for figure 4:

**Figure supplement 1.** Evolution of cross-covariance between movement speed and PV activity.

to movement peaks, no elevated activity was observed (*Figure 3—figure supplement 2C*). We varied the overhead lighting and found that IL PV neurons showed no elevated activity during movement under either a bright ceiling light, which is a more aversive setting to the mice, or a dim ceiling light, which is more comforting to the mice (*Figure 3—figure supplement 2D and E*, N=7 mice, bright light: p=0.8705, dim light: p=0.1475, paired t-test; *Figure 3—figure supplement 2F*). It should be noted that the OFT is a relatively neutral context, regardless of lighting conditions, when compared to the appetitive reward approach or aversive active avoidance.

## IL PV neural activity becomes positively correlated to movement after shock

In environments with different emotional valence animals engage in different suites of behaviors. For example, animals in threatening environments spend more time freezing and less time exploring than animals in rewarding environments. We hypothesized that IL neural activity may become positively correlated with movement when animals learn that their environment is threatening, and must suppress behaviors such as freezing in order to engage in behaviors such as exploration.

To test this hypothesis, we analyzed the recordings made on the first day of active avoidance training to determine how movement-related IL PV neural activity evolves as animals learn that the environment is threatening. On the first session, we habituated animals to the chamber for 5 min before the task started. During this habituation period, IL PV activity was not positively correlated with movement (*Figure 4A*; *Figure 4—figure supplement 1A*), similar to our observations in appetitive and neutral contexts (*Figure 3*; *Figure 3—figure supplement 1E–I*). During avoidance behavior, we found that IL PV activity peaked during movements in the inter-trial interval (*Figure 4B*; *Figure 4—figure*

*supplement 1A*). Considering that animals minimize their movements in threatening environments, movement outside of the tone-evoked trial could require suppression of this natural inclination.

The correlation between movement and IL PV neural activity during the inter-trial interval did not emerge until after animals experienced the first foot shock (*Figure 4C*; *Figure 4—figure supplement 1A*). The correlation between movement and IL PV neuronal activity emerged immediately after receiving the first shock and did not increase during the learning process, suggesting that avoidance learning is not necessary for the correlation between movement and IL PV neural activity to emerge. The lack of positive correlation between activity and movement in the habituation phase (*Figure 4A–C*; *Figure 4—figure supplement 1A*) of the avoidance task is similar to what was observed throughout the approach task (*Figure 4D*; *Figure 4—figure supplement 1B*). These data show that the positive correlation between movement and PV neuron activity emerges immediately after animals learn that their current environment is threatening, when movement requires a suppression of freezing.

## Inhibiting IL PV neural activity delays avoidance

To investigate the causal role of IL PV neural activity in active avoidance, we bilaterally inhibited IL PV neurons by expressing Cre-dependent halorhodopsin (AAV5-EF1α-DIO-eNpHR; control animals: AAV5-EF1α-DIO-eYFP) in PV-Cre mice (*Figure 5A and B*). We used the same avoidance and approach tasks described above (*Figure 1A*; *Figure 3A*), and introduced interleaved simulation blocks where IL PV neurons were optogenetically inhibited during the interval from 0.5 to 2.5 s after tone onset (*Figure 5C*). Inhibiting IL PV neuronal activity delayed the avoidance movement (*Figure 5D–G*; *Figure 5—figure supplement 1A and B*; *Video 1*) but did not delay the movement for water reward in the approach task (*Figure 5H–K*; *Figure 5—figure supplement 1C and D*) or the speed of voluntary locomotion in the OFT (*Figure 5L–O*, dim OFT: interaction between opsins and stimulation: N=8 NpHR mice, N=5 eYFP mice, p=0.7636, two-way ANOVA; bright OFT: interaction between opsins and stimulation: N=8 NpHR mice, N=5 eYFP mice, p=0.7469, two-way ANOVA).

Both the speed of the avoidance movement (*Figure 5D and E*, significant interaction between opsin and stimulation, N=8 NpHR mice, N=5 eYFP mice, p=0.0281, two-way ANOVA) and the probability of successful avoidance (*Figure 5F and G*, N=8 NpHR mice, N=5 eYFP mice, p=0.0058, unpaired t-test) were reduced by suppression of IL PV neurons. The speed of movement and the probability of successful reward delivery were not affected in the reward approach task (speed: *Figure 5H and I*, interaction between opsin and stimulation, N=8 NpHR mice, N=5 eYFP mice, p=0.479, two-way ANOVA; probability of approach: *Figure 5J and K*, N=8 NpHR mice, N=5 eYFP mice, p=0.1204, unpaired t-test). This suggests that IL PV neuronal activity is not just correlated with avoidance behavior but plays a causal role. By further analyzing the video data, we found that freezing during tone presentation increased (*Figure 5—figure supplement 1E–G, G*: N=8 NpHR mice, N=5 eYFP mice, p=0.0175, unpaired t-test; *Video 1*). In addition, the ratio of freezing duration to avoidance latency increased during IL PV inhibition (*Figure 5—figure supplement 1H*, N=8 NpHR mice, N=5 eYFP mice, p=0.0302, unpaired t-test; *Video 1*), suggesting that increased avoidance latency was primarily due to an increase in freezing. This finding supports the idea that IL PV neural activity promotes avoidance by suppressing freezing (*Murphy et al., 2005*; *Hardung et al., 2017*).

We then asked whether inhibiting IL PV neuronal activity had an immediate effect on avoidance during the current trial or if instead inhibition influenced learning. To address this question, we investigated whether IL PV inhibition affected avoidance in trials following inhibition, using the task described in *Figure 5C*. Contrary to the learning hypothesis, we found that suppressing IL PV neurons delayed avoidance even on the first trial of a stimulation block. The latency of the first trial of inhibition blocks was significantly longer than the latency of pre-stimulation trials (*Figure 5—figure supplement 1I and J*, N=8 NpHR mice, N=5 eYFP mice, p=0.0022, paired t test), but was not significantly different from the latency of subsequent inhibition trials (*Figure 5—figure supplement 1I and J*, N=8 NpHR mice, N=5 eYFP mice, p=0.0943, paired t test). The latency of the first non-inhibited trial following an inhibition block is significantly different from the latency of the preceding inhibition trials (*Figure 5—figure supplement 1K anf L*, N=8 NpHR mice, N=5 eYFP mice, p=0.0343, paired t test) but is not significantly different from the latency of subsequent non-inhibited trials (*Figure 5—figure supplement 1K and L*, N=8 NpHR mice, N=5 eYFP mice, p=0.8147, paired t-test). These results suggest that IL PV neuronal activity plays a causal role in modulating ongoing behavior.

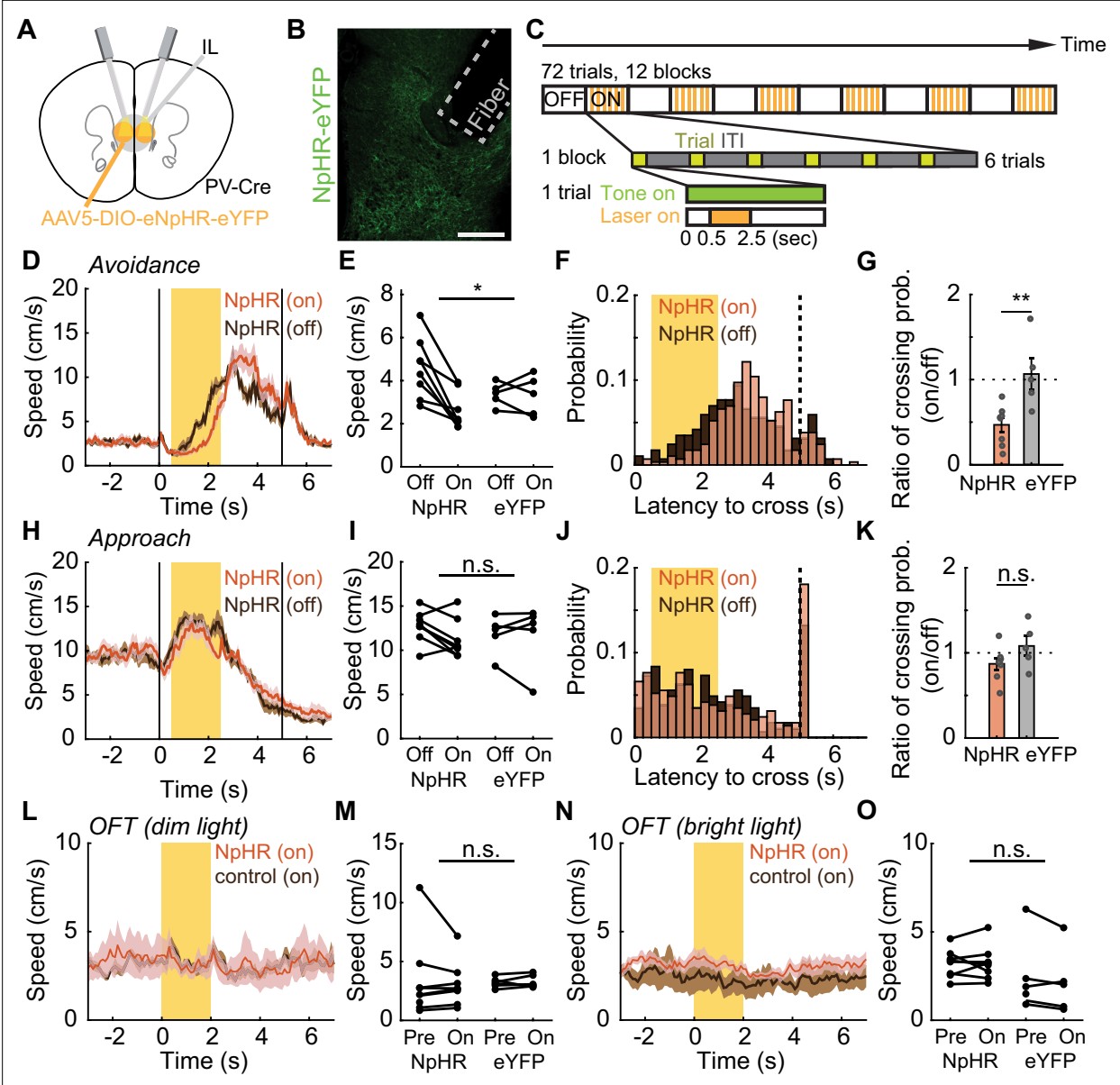

**Figure 5.** Inhibiting IL PV neural activity delays avoidance. (**A**) Optogenetic schematic. (**B**) NpHR-eYFP expression in IL PV neurons in a PV-Cre mouse. Scale bar: 200 μm. (**C**) Optogenetic inactivation schematic. Interleaved simulation blocks were introduced, with optogenetic inhibition of IL PV neurons 0.5–2.5 s after tone onset. (**D**) Speed of NpHR-expressing mice during avoidance with (red) and without (dark brown) illumination. (**E**) Speed of NpHR- and eYFP-expressing mice, averaged over the laser stimulation period, with and without illumination. (**F**) Distribution of crossing latencies during avoidance in NpHR-expressing animals with (red) and without illumination (brown). (**G**) Ratio of illuminated/non-illuminated chamber crossing probabilities. (**H–K**) Same as (**D–G**) for approach task. (**L–M**) Same as (**D–E**) but for OFT with dim light. (**N–O**) Same as (**L–M**) but for OFT with bright light. ns = nonsignificant, *p<0.05, **p<0.01; for (**E**), (**I**), (**M**), and (**O**), two-way ANOVA interaction term; for (**G**) and (**K**), unpaired t-test. Shaded regions indicate SEM.

The online version of this article includes the following figure supplement(s) for figure 5:

**Figure supplement 1.** Controls for IL PV inhibition.

To further tease apart the role of IL PV neural activity in learning, we inhibited IL PV neurons immediately after successful chamber crossing in the avoidance task, rather than during the tone (*Figure 5—figure supplement 1M*). With this experimental design, speed was not affected by IL PV inhibition (*Figure 5—figure supplement 1N–P, P*: N=8 NpHR mice, N=5 eYFP mice, p=0.8846, two-way ANOVA interaction term). The latency of the first trials in inhibition blocks showed no significant difference from the latency of the second trials in inhibition blocks (*Figure 5—figure supplement*

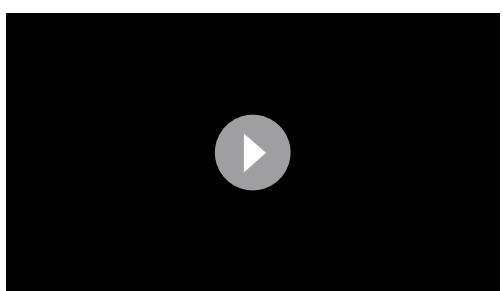

**Video 1.** Inhibiting IL PV neuronal activity delays the avoidance movement.

https://elifesciences.org/articles/91221/figures#video1

*1Q*, N=8 NpHR mice, N=5 eYFP mice, p=0.6944, paired t-test), and the latency of the first trials after the end of the inhibition block also showed no difference to the latency of the second trials after the inhibition block (*Figure 5—figure supplement 1Q*, N=8 NpHR mice, N=5 eYFP mice, p=0.6745, paired t-test). Thus, post-avoidance inhibition of IL PV neural activity did not affect avoidance latency in subsequent trials.

## Discussion

Here, we show that IL PV neural activity rises during movement to avoid a cued future shock, but not during movement to obtain reward or movement in the open field. Further, this rise in activity during movement emerges immediately after the first shock and does not require avoidance learning. We also show that inhibiting IL PV neurons prolongs freezing and delays avoidance, but does not affect movement to obtain reward or movement in the open field. These results demonstrate that IL PV neurons play a key role in suppressing freezing in order to permit movement in threatening environments, an essential component of the avoidance response.

The rise in IL PV neural activity during movement does not require avoidance learning – IL PV neurons begin to respond during movement immediately after the animal has received a single shock in an environment, but learning to cross the chamber to avoid the signaled shock takes tens of trials. Why is there a discordance between the emergence of the IL PV signal during movement and avoidance learning? The components underlying active avoidance have been debated over the years, but are thought to involve at least two essential behaviors – suppressing freezing, and moving to safety (*LeDoux et al., 2017*). Freezing is the default response of mice upon hearing a shock-predicting tone, and can be learned in a single trial (*Ledoux, 1996*; *Fanselow, 2010*; *Zambetti et al., 2022*). When a predator is in the distance, freezing can increase the chance of survival by reducing the chances of detection. However, a strategic avoidance behavior may prevent a future encounter with the predator altogether. The importance of IL PV neural activity in defensive behavior may be to suppress reactive defensive behaviors such as freezing in order to permit a flexible goal-directed response to threat.

The freezing suppression and avoidance movement components of the avoidance response are dissociable, both because freezing precedes avoidance learning, and because animals intermittently move prior to avoidance learning. Our finding that the rise in PV activity during movement emerges immediately after receiving a single shock, tens of trials before animals have learned the avoidance behavior, suggests that the IL PV signal is associated with the suppression of freezing. Further, IL PV neurons do not respond during movement toward cued rewards because in reward-based tasks there is no freezing response in conflict with reward approach behavior.

We think the IL PV signal is unlikely to be a safety signal (*Sangha et al., 2020*). First, the PV signal rises during movement not only in the avoidance context, but during any movement in a 'threatening' context (i.e. a context where the animal has been shocked). For example, PV neural activity rises during movement during the intertrial interval in the avoidance task. Further, the emergence of the PV signal during movement happens quickly – after the first shock – and significantly before the animal has learned to move to the safe zone. This suggests a close association with enabling movement in a threatening environment, when animals must suppress a freezing response in order to move. Additionally, the rise in PV activity was specifically associated with movement and not with tone offset, the indicator of safety in this task. Finally, if IL PV neural activity reflects safety signals one would expect the response to be enhanced by learning, but the amplitude of the IL PV response was unaffected by learning after the first shock.

Finding that inhibitory IL PV neural activity suppresses freezing was counterintuitive, given the importance of IL for fear extinction and avoidance learning. We had predicted that IL PV neurons would be suppressed during avoidance, because IL is active when animals have successfully extinguished freezing in response to shock-predicting cues (*Milad and Quirk, 2002*), and muscimol inhibition of IL impairs avoidance learning (*Moscarello and LeDoux, 2013*). However, recent studies have

suggested that the role of cortical PV neurons goes beyond suppressing overall neural activity in a region, and likely plays a more delicate role in tuning local computations.

For example, PV neurons aid in improving visual discrimination through sharpening response selectivity in visual cortex (*Lee et al., 2012*). In prefrontal cortex, PV neurons are critical for task performance, particularly during performance of tasks that require flexible behavior such as rule shift learning (*Cho et al., 2020*) and reward extinction (*Sparta et al., 2014*). Further, PV neurons play an essential role in the generation of cortical gamma rhythms, which contribute to synchronization of selective populations of pyramidal neurons (*Sohal et al., 2009*; *Cardin et al., 2009*). *Courtin et al., 2014* showed that brief suppression of dorsomedial prefrontal (dmPFC) PV neural activity enhanced fear expression, one of the main functions of the dmPFC, by synchronizing the spiking activity of dmPFC pyramidal neurons (*Courtin et al., 2014*). This result is potentially relevant to our findings, but likely involves different circuit mechanisms because of the difference in timescale, targeted area, and downstream projection targets (*Vertes, 2004*). These and other studies support the idea that PV neural activity supports the execution of a behavior by shaping rather than suppressing cortical activity, potentially by selecting among conflicting behaviors by the synchronization of different pyramidal populations (*Warden et al., 2012*; *Lee et al., 2014*). The roles of other inhibitory neural subtypes (such as somatostatin (SOM)-expressing and vasoactive intestinal peptide (VIP)-expressing IL GABA neurons) in avoidance behavior are currently unknown, but are likely important given the role of SOM neurons in gamma-band synchronization (*Veit et al., 2017*), and the role of VIP neurons in regulating PV and SOM neural activity (*Cardin, 2018*).

Our findings have revealed the role of IL PV neural activity in facilitating flexible avoidance behavior by suppressing the conflicting freezing behavior. Though IL PV neurons comprise only a relatively sparse cortical population, inhibiting these neurons has a clear and specific detrimental effect on the initiation of avoidance behavior, which is vital for leaving a dangerous situation. Our work also suggests that the functional role of PV neurons extends beyond the overall suppression of the function of a brain region, and provides a conceptual framework of potential utility for deepening our understanding of the functional roles of PV neurons in mediating conflict between behaviors through coordination of local circuits.

# Materials and methods

## Animals

All procedures conformed to guidelines established by the National Institutes of Health and have been approved by the Cornell University Institutional Animal Care and Use Committee. PV-Cre mouse line (B6.Cg-*Pvalb*^tm1.1(cre)Aibs/J, RRID:IMSR_JAX:012358) acquired from The Jackson Laboratory (Bar Harbor, ME) was backcrossed to C57BL/6 J mice (RRID:IMSR_JAX:000664). Postnatal six weeks to ten months old PV-Cre male mice were used for all photometry and optogenetics experiments. All mice were housed in a group of two to five under 12 hr reverse light-dark cycle (dark cycle: 9 a.m.-9 p.m.).

## Viral vectors

In the photometry experiment, we used AAV1-CAG-Flex-GCaMP6f (Titer: $1.33\times10^{13}$, Penn Vector Core, 100835-AAV1, Philadelphia, PA) for experimental animals and AAV9-CAG.Flex.GFP (Titer: $3.7\times10^{12}$, UNC Vector Core, Chapel Hill, NC) for control. All the viral vectors used in photometry experiment were diluted in eightfold PBS before injection. In the optogenetics experiment, we used AAV5-EF1α-DIO-eNPHR3.0 (Titer: $4\times10^{12}$, UNC Vector Core, Chapel Hill, NC) for experimental animals and AAV5-EF1α-DIO-EYFP (Titer: $6.5\times10^{12}$, UNC Vector Core, Chapel Hill, NC) for control. All the viral vectors used in optogenetics experiment were used without dilution.

## Surgical procedures

Mice were put under deep anesthesia with isoflurane (5%). Fur above the skull was trimmed, and the mice were placed in a stereotaxic frame (Kopf Instrument, Tujunga, CA) with a heating pad to prevent hypothermia. Isoflurane level was kept between 0.8 to 2% throughout the surgery. Ophthalmic ointment was applied to protect the eyes. 100 µL 1 mg/ml Baytril (enrofloxacin) was given subcutaneously, and 100 µL 2.5 mg/ml bupivacaine was injected subdermally at the incision site. The scalp was

disinfected with betadine and alcohol. The skull was exposed with a midline incision. A craniotomy was made above the medial prefrontal (mPFC) cortex.

For fiber photometry animals, virus (AAV1-CAG-Flex-GCAMP6f) was injected into mPFC unilaterally, with half of the animals into the right hemisphere and half of the animals into the left hemisphere (infralimbic cortex (IL) coordinates: –1.55 AP,±0.3 ML, –2.8 to –3.2 DV). A total of 800 nl diluted vector (1:8 dilution) was injected in each mouse. Virus injection was done with a 10 µL Hamilton syringe (nanofil, WPI, Sarasota, FL) and a 33-gauge beveled needle, and a micro-syringe pump controller (Micro 4; WPI, Sarasota, FL) using slow injection speed (100 nl/min). The needle was slowly withdrawn 15 min after injection. After injection, a 4 mm or 6 mm-long optic fiber (diameter: 400 µm, 0.48NA, Doric Lenses, Quebec, Canada) was implanted 0.5–1 mm above the injection site.

For optogenetics animals, virus (AAV5-EF1a-DIO-eNpHR3.0, Lot #: 4806 G, titer: $4.00 \times 10^{12}$; control: AAV5-EF1a-DIO-eYFP, Lot #: 4310 J, titer: $6.5 \times 10^{12}$, UNC vector core, NC, USA) was injected into mPFC bilaterally (IL coordinates: –1.55 AP,±0.3 ML, –2.8 to –3.2 DV). 500 nl vector was injected into each site on each mouse. After injection, a 4 mm to 6 mm-long optic fiber (diameter: 200 µm, 0.22NA, Thorlabs, NJ, USA) was implanted 0.5–1 mm above the injection site in a 15 degree angle toward midline (AP = 1.55, ML = ±1.05 or 1.15, DV = −2.8 or −2.99, 15 degree angle).

After the fiber implant, a layer of metabond (Parkell, Inc, Edgewood, NY) and dental acrylic (Lang Dental Manufacturing, Wheeling, IL) was applied to hold the implant in place. Buprenorphine (0.05 mg/kg), carprofen (5 mg/kg), and lactated ringers (500 µL) were administered subcutaneously after surgery. Photometry recording was done no earlier than three weeks later to allow for virus expression.

## Fiber photometry

Fiber photometry was implemented with a fiber photometry console (Doric Lenses, Quebec, Canada). An FC-FC optic fiber patch cord (400 µm diameter, 0.48NA, Doric Lenses, Quebec, Canada) was connected to implanted fiber with a zirconia sleeve. 405 nm and 475 nm were measured for calcium-independent and calcium-dependent GCaMP signals and were measured with digital lock-in frequency to the input at 208 Hz and 530 Hz, respectively. Photometry signals were collected at 12 kHz and filtered with a 12 Hz low pass filter.

For photometry animals, four GCamp6f animals were tested only on active avoidance. Seven GCamp6f animals and two GFP control animals were tested in the following order: open field test, reward approach test, reward approach with omission (except two GFP and two GCamp6f animals), active avoidance, active avoidance with shortening and extended tone, active avoidance extinction.

### Open field test (photometry)

A 46 W x 46 L x 30.5 H (cm) white rectangular box made with PVC was used for an open field test. The ambient light was set for each 2 minute block in the order of D(dim)-B(bright)-D-B-D-B-D-B-D-B.

### Reward approach (photometry)

The reward approach task was performed in a 17.25" W x 6.75" D x 10" H metal rectangular shuttle box (MedAssociates, Fairfax, VT) divided into two equal compartments by Plexiglas semi-partitions, which allowed animals to move freely between compartments. A water sprout was located at the end of each compartment, and a syringe pump was connected to the sprout for water delivery. Licks were detected with a contact-based lickometer (MedAssociates, Fairfax, VT). Animals were water restricted prior to training. Body weight was checked daily and was maintained above 80% baseline. Animals were trained to first learn the association between tone and reward-licking at the waterspout in the opposite compartment, by playing a tone for an indefinite length of time (12 kHz or 8 kHz at 70–80 dB, counterbalancing between approach and avoidance) until successful reward collection. When an animal crossed the chamber in response to a tone, water was delivered in the goal compartment, and an indicator light above the targeted water sprout was terminated. After two to three days of training, the reward-indicating lights were removed. Tone duration was set to be turned off either at the chamber crossing or at maximum duration. Animals had to cross within the maximum duration of tone for successful water delivery. The maximum tone duration was shortened as animals progressed and were eventually set to 5 s. The inter-trial interval was pseudo-randomized at an average of 40 s. Animals were allowed to perform 30–50 trials each day during training and 100 trials on the recording

day. The training lasted two to three weeks until animals reached a 70% success rate with a 5 s window. Animals were recorded the day after criterion was reached for 100 trials.

### Reward approach with 10 % omission (photometry)

After training and recording in reward approach, animals were then switched to a 10% omission paradigm, where water was not delivered on 10% of the successful crossing trials. The omission trials were selected pseudo-randomly at a 10% chance.

### Active avoidance (photometry)

Active avoidance was performed in a 14" W x 7" D x 12" H metal rectangular shuttle box (Coulbourn Instruments, Holliston, MA) divided into two equal compartments by Plexiglas semi-partitions. Animals were habituated to the chamber for at least 30 min, the day before training. On the first day, animals received five 5-s habituation tones and two Pavlovian conditionings (a 7-s tone and a 2-s shock at the last 2-s of tone) prior to avoidance trials. The same frequency and amplitude of tone were used for habituation tones, Pavlovian tones, and avoidance tones (12 kHz or 8 kHz at 70–80 dB, counterbalancing between approach and avoidance). After habituation and Pavlovian trials, the task was switched to avoidance trials where animals could prevent the shock by crossing the chamber within 5 s from the onset of the tone. Otherwise, an electrical foot shock (0.3 mA) would be delivered through the grid floor for a 2 s maximum before the animal crossed the chamber to escape the shock. The tone was terminated when animals crossed the chamber or after 7 s. The inter-trial interval was pseudo-randomized for an average of 40 s. Animals performed 100 avoidance trials per day for 2–3 days or until reaching 70% successful rate. Photometry data was recorded during training.

### Active avoidance with 10% shortened and extended tones (photometry)

After active avoidance recording, animals were then recorded in an alternative version of active avoidance with 10% shortened and extended tones. The only change in this alternative version was to include 10% trials with shortened tones which were turned off after 1.5 s regardless, and 10% trials with extended tones, which were extended for 1.5 s after successful avoidance. The shortened and extended trials were selected pseudo-randomly at 10% chance for each trial. No shock was delivered in any shortened-tone and extended-tone trials.

## Optogenetics

In the behavioral experiment, two external FC-FC optic fiber patch cords (200 μm diameter, 0.22 NA, Doric Lenses, Quebec, Canada) were connected to two implanted fibers, respectively, each with a zirconia sleeve. These patch cords were then connected to a 1x2 fiber-optic rotary joint (FRJ_1x2i_FC-2FC_0.22, Doric Lenses, Quebec, Canada) for unrestricted rotation and to prevent tangling. Another FC-FC optic fiber patch cord was used to connect the rotary joint to a 100 mW 594 nm diode-pumped solid-state laser (Cobolt Mambo 100 594 nm, HÜBNER Photonics, Sweden) for optogenetic stimulation. The power of the laser was programmed by the software and fine-tuned by a continuous filter (NDC-50C-2M, Thorlabs, NJ, USA) to 10 mW at the end of the patch cord (~71.59 mW/mm$^2$ at the end of the implanted fiber). The stimulation timing was controlled by a shutter (SRS470, Stanford Research System, Sunnyvale, CA) and a Master-8 stimulus generator (A.M.P.I., Jerusalem, Israel). In avoidance and approach, a total of 72 trials were divided into 12 alternating blocks (OFF-ON-OFF…), and 2 s continuous stimulation was delivered 0.5 s after the onset of a tone during stimulation blocks. In the open field test, a 32 min test was divided into eight alternating stimulation blocks (OFF-ON-OFF…), and a 2 s continuous stimulation was delivered every 40 s during the stimulation blocks.

For optogenetics animals, eight experimental and five controls were tested in the following order: open field test under dim light, reward approach, active avoidance, open field test under bright light.

### Reward approach (optogenetics)

Animals were trained using the same conditioning and chamber and tested in the same chamber as mentioned in *Reward Approach (Photometry)*. After animals reached the learning criteria (70% success rate with 5 s window), animals were then trained with a patch cord attached for another 1–2 days to habituate the animals to a patch cord. On the test day, 72 trials were divided into 12

alternating blocks (OFF-ON-OFF……), and a 2 s continuous stimulation was delivered 0.5 s after the onset of tone during stimulation blocks, regardless of the behavioral outcome.

### Active avoidance (optogenetics)

Animals were trained using the same conditioning and chamber, and were tested in the same chamber as mentioned in *Behavior Paradigm: Active Avoidance (Photometry)*. After animals reached the learning criteria (70% success rate with 5 s window), animals were then trained with a patch cord attached for another 1–2 days to habituate the animals to a patch cord. On the test day, 72 trials were divided into 12 alternating blocks (OFF-ON-OFF……), and a 2 s continuous stimulation was delivered 0.5 s after the onset of tone during stimulation blocks, regardless of the behavioral outcome.

### Open field test (optogenetics, dim light)

A 26 W x 48 L x 21 H (cm) clean rectangular rat homecage with mouse homecage bedding placed in a sound-proof box (MedAssociates, Fairfax, VT) lit by a red LED strip was used for an open field test. Mice were first habituated with their cagemates, food, and water in the arena for an hour the day before testing. During the test, food and water were removed from the arena, and each mouse was tested individually. At the start of the experiment, mice were first connected to the patch cord fiber and then placed in the center of the arena. A 32-min test was divided into eight alternating stimulation blocks (OFF-ON-OFF……), a 2-s continuous stimulation was delivered every 40 s during the stimulation blocks.

### Open field test (optogenetics, bright light)

A 46 W x 46 L x 30.5 H (cm) white rectangular box made with PVC was used for an open field test under bright room light. Mice were first connected to the patch cord fiber and then placed in the center of the arena at the start of the experiment. A 32-min test was divided into eight alternating stimulation blocks (OFF-ON-OFF……), a 2-s continuous stimulation was delivered every 40 s during the stimulation blocks.

## Perfusion and histology verification

After experiments, animals were deeply anesthetized with pentobarbital at a dose of 90 mg/kg and perfused with 20 ml PBS, followed by 20 ml 4% paraformaldehyde solution. Brains were soaked in 4 °C 4% paraformaldehyde for 20 hr and then switched to 30% sucrose solution for 20–40 hr until the brains sank. Brains were sectioned coronally (40–50 µm) with a freezing microtome and then washed with PBS and mounted with PVA-DABCO. Images were acquired using a Zeiss confocal with 5 x air, 20 x water, and 40 x water objectives.

## Statistics and data analysis

All data analysis and statistical testing were performed using custom-written scripts in MATLAB 2019 (MathWorks, Natick, MA). For all behaviors, location and movement were tracked using Ethovision XT10 (Noldus Leesburg, VA).

Error bars and shaded areas in figures report standard error of the mean (s.e.m.). All statistical tests were two-tailed. Within-subject analyses were performed using paired t-test, and between-subject analyses were performed using an unpaired t-test.

## Photometry signal analysis

ΔF/F was calculated with equation (*Equation 1*). The signal measured from the 405 nm reference channel was linearly fitted to the 475 nm signal and was subtracted from the 475 nm signal. The difference was then divided by an exponential function ($a \cdot e^{-b} + c$) fitted to 475 nm signal.

$$\frac{\Delta F}{F} = \frac{475nmsignal - fitted405nmsignal}{exponentialdecayfittedto475nmsignal} \times 100 \qquad (1)$$

## Open field test analysis

Mouse speed was first grouped into three clusters, nonmovement, low movement, and high movement, using k-means. The movement threshold was thus defined by the lowest speed of the low movement cluster. Then the peak speed of the movement was detected by finding local maxima with absolute peak value larger than a threshold and at least 5 cm/s larger than the baseline. (This was performed using *findpeak* function in MATLAB and with 'MinPeakProminence' set to 5 and with 'MinPeakHeight' set to movement threshold.) Movement initiation was defined by when the speed first went above 10% of the peak speed within 2 s before the epoch. Only epochs with at least 1 s of less than threshold speed before movement initiation were included. Group photometry analyses compared mean ΔF/F 4–2 s before and 2 s around the peak speed of the movement. Optogenetic analyses compared speed differences of NpHR- and eYFP-expressing mice between mean speed during the 2-s stimulation period and mean speed at the 2-s period right before stimulation.

## Approach task and active avoidance task

### Group photometry analysis

Group photometry analyses compared mean ΔF/F 4–2 s before and 1 s before and after chamber crossing, and only successful trials where animals crossed the chamber within a 5-s window were included. In the active avoidance task, movement initiation was defined by when the speed first went above 10% of the speed at the chamber crossing within 2 s before the chamber crossing. In *Figure 1—figure supplement 1F, G and H*, the maximal ΔF/F around crossing was calculated by taking the maximum ΔF/F from 0.5 s before to 1 s after avoidance chamber crossing of each trial, and the corresponding maximal speed around crossing was calculated by taking the maximum speed 0.5 s before and after avoidance chamber crossing.

### Movement detection

Movement epochs in habituation and inter-trial intervals in avoidance tasks were detected by finding local maxima with absolute peak value larger than 10 cm/s and at least 3 cm/s larger than the baseline in smoothed speed traces. This was performed using *findpeak* function in MATLAB and with 'MinPeakProminence' set to 3 and with 'MinPeakHeight' set to 10. The speed traces were smoothed with Gaussian-weighted moving average with a window of seven frames by MATLAB function *smoothdata* before the movement epoch detection. (The video was recorded in 15 frames per second). The movement was excluded if it happened within 5 s after another movement. Habituation was defined as the time starting from when the animals were placed in the chamber to the start of the first Pavlovian conditioning trials on the first day of training.

Movement in the intertrial intervals was measured from 5 s after the end of the trial to 5 s before the start of the next trial in a well-learned session. Movement initiation was defined by when the speed first went above 10% of the peak speed within 2 s before the epoch.

### Group optogenetic speed and crossing probabilities analysis

Optogenetic speed analyses compared speed differences of NpHR- and eYFP-expressing mice during the 2-s stimulation period (0.5 s to 2.5 s from the tone onset) in stimulation trials to the same period in non-stimulation trials. Trials in which animals crossed before 0.5 s after tone onset or before the laser stimulation onset were excluded from speed analyses (*Figure 5D-E, H-I*, *Figure 5—figure supplement 1A and C*). To plot a histogram of crossing probabilities under optogenetic stimulation, we first binned the crossing latency of all trials into 0.25 s bins ranging from 0 to 7 s and then normalized by the total number of trials to obtain crossing probability. Optogenetic crossing probability analyses compared differences between NpHR- and eYFP-expressing mice in crossing probability during the 2-s stimulation period (0.5 s to 2.5 s from the tone onset) in stimulation trials to non-stimulation trials.

### Group optogenetic block structure analysis

In *Figure 5—figure supplement 1I—L*, total of 72 active avoidance trials with 12 off-on alternating stimulation blocks, pattern described in *Figure 5A*, were sectioned into six chunks, each chunk containing six trials before stimulation and six trials with stimulation. Each data point at each time point is the average latency across six chunks of all NpHR-expressing mice. In *Figure 5—figure*

*supplement 1K—L*, a total of 72 active avoidance trials with 12 off-on alternating stimulation blocks, pattern described in *Figure 5A*, were then sectioned into five chunks, each chunk containing six trials with stimulation and six trials after stimulation. Due to the stimulation pattern, first 'off' block and last 'on' blocks were not included. Each data point at each time point is the average latency across five chunks of all NpHR-expressing mice. In *Figure 5—figure supplement 1Q*, a total of 72 active avoidance trials were sectioned into five chunks; each chunk contains three trials before stimulation, six trials with stimulation, and three trials after stimulation. Each point plotted on the graph is the average across five chunks and of all NpHR-expressing animals.

## Freezing detection

Freezing is detected from video based on changes in pixels by a MATLAB function written by David A. Bulkin and Ryan J. Post. To detect the freezing of animals, we used the code first to convert the video into grayscale, crop the window to include only the bottom of the chamber, and then set the pixels belonging to a mouse to 1 s by thresholding pixel values below 27 out of 255. Once the pixels belonging to a mouse were assigned, the code compared the number of pixels changed between frames to get raw movement data. Raw movement data was then filtered with a bandpass filter between 0.01 and 0.9 to ensure the frequency of switching between freezing and non-freezing states matches with the behavior observed. Freezing was then marked when less than 190 mouse pixels were changed in filtered movement data. For *Figure 5—figure supplement 1H*, duration of freezing was calculated by the total number of frames when the animals were detected freezing during avoidance latency.

## Cross-covariance between photometry signals and speed

Photometry signals were first filtered by a lowpass IIR filter with half-power frequency at 7 Hz and then interpolated to match video recording timeframes (15 Hz). Each segment included photometry and speed data from 5 s after tone offset to 5 s before the next tone onset. Cross-covariance was calculated with an offset of ±2 s for each segment by the *xcov* function in MATLAB. The cross-covariance value with the largest absolute value within the offset range was selected to be the cross-covariance value of the segment (*Seo et al., 2019*).

## Linear regression model

The calcium signal was modeled as a linear combination of characteristic neural responses to each sensory or action event following the scheme of *Musall et al., 2019* and *Parker et al., 2016* (*Musall et al., 2019*; *Parker et al., 2016*). These events included tone onset, tone offset, avoidance, escape, shock onset, and chamber crossing in the intertrial intervals. The model can be written as the following equation, where ΔF/F (t) is the recorded calcium dynamics at time t, $k_i$ and $j_i$ are the kernel coefficients of event type I, $\tau$ is the relative time points in a kernel, n1 is the total number of action events, and n2 is the total number of sensory events. Each event kernel $k_a$ or $j_b$ was placed at the time point where the event a or b occurred. For action events, including avoidance, escape, and chamber crossing in intertrial intervals, we used kernels ($k_i$) ranging from 1 s before to 2 s after the event to cover the action from initiation to termination. For sensory events, including tone onsets, tone offsets, and shock onsets, kernels ($j_i$) started right at the moment where the event happened and lasted for 2 s afterwards to model the sensory responses.

$$\Delta F/F\left(t\right) = \sum_{i=1}^{n1}\left(\int_{\tau=-1s}^{2s} k_i\left(\tau\right) * \left(t-\tau\right) d\tau\right) + \sum_{i=1}^{n2}\left(\int_{\tau=0s}^{2s} j_i\left(\tau\right) * \left(t-\tau\right) d\tau\right) + error \qquad (2)$$

ΔF/F (t) was modeled as the linear combination of all event kernels. Given the event occurrence time points of all event types, we can use linear regression to decompose characteristic kernels for each event type. Kernel coefficients of the model were solved by minimizing the mean square errors between the model and the actual recorded signals. To prove that kernel $k_i$ is an essential component for the raw calcium dynamics, we compared the explanation power of the full model to the reduced model where the time points of the occurrence of event $k_i$ were randomly assigned. Thus, the kernel coefficients should not reflect the response to the event in the reduced model. The coefficients of determination ($R^2$) were compared between the reduced model and the full model to estimate the

unique contribution of certain events to the explanation power of the model. More details of the methods can be viewed in *Musall et al., 2019*; *Parker et al., 2016*.

## Acknowledgements

We thank JR Fetcho, RM Harris-Warrick, HK Reeve, JH Goldberg, N Yapaci, Y Baumel, W-K You, O Gschwend, W-S Wei, W-C Huang, J Cia, T Zhou, and W Menegas for helpful discussion, BJ Sleezer, E Troconis, A Guru, RJ Post and C Seo for assistance with fiber photometry and behavior, and Y Baumel, C Seo and BJ Sleezer for advice on data analysis. This work was supported by the Mong Family Foundation (YH), the Taiwan Ministry of Education (YH), NIH DP2MH109982 (MRW), the New York Stem Cell Foundation (MRW), the Alfred P Sloan Foundation (MRW), the Whitehall Foundation (MRW), and the Brain and Behavior Research Foundation (MRW).

## Additional information

### Funding

| Funder | Grant reference number | Author |
| --- | --- | --- |
| NIH Director's New Innovator Award | DP2MH109982 | Melissa R Warden |
| New York Stem Cell Foundation | Robertson Neuroscience Investigator Award | Melissa R Warden |
| Whitehall Foundation | Research Grant | Melissa R Warden |
| Alfred P. Sloan Foundation | Sloan Research Fellowship | Melissa R Warden |
| Brain and Behavior Research Foundation | NARSAD Young Investigator Award | Melissa R Warden |
| Mong Family Foundation | | Yi-Yun Ho |
| Taiwan Ministry of Education | | Yi-Yun Ho |

The funders had no role in study design, data collection and interpretation, or the decision to submit the work for publication.

### Author contributions

Yi-Yun Ho, Conceptualization, Formal analysis, Funding acquisition, Investigation, Writing – original draft, Writing – review and editing; Qiuwei Yang, Investigation, Writing – review and editing; Priyanka Boddu, Formal analysis, Investigation, Writing – review and editing; David A Bulkin, Formal analysis, Writing – original draft, Writing – review and editing; Melissa R Warden, Conceptualization, Formal analysis, Supervision, Funding acquisition, Writing – original draft, Project administration, Writing – review and editing

### Author ORCIDs

Yi-Yun Ho ⬤ https://orcid.org/0000-0002-2465-790X
Melissa R Warden ⬤ https://orcid.org/0000-0003-2240-3997

### Ethics

All procedures conformed to guidelines established by the National Institutes of Health and have been approved by the Cornell University Institutional Animal Care and Use Committee.

Reviewer #2 (Public Review): https://doi.org/10.7554/eLife.91221.3.sa1
Reviewer #3 (Public Review): https://doi.org/10.7554/eLife.91221.3.sa2
Author response https://doi.org/10.7554/eLife.91221.3.sa3

## Additional files

### Supplementary files
MDAR checklist

### Data availability
Custom MATLAB data analysis code is publicly available on GitHub (copy archived at *Ho, 2025*). Source data used for statistical analysis and to create the figures are publicly available at Open Science Framework. Warden, M., and Ho, Y.-Y. (2025). Infralimbic parvalbumin neural activity facilitates cued threat avoidance. Available at: osf.io/679jy.osf.io/679jy.

The following dataset was generated:

| Author(s) | Year | Dataset title | Dataset URL | Database and Identifier |
|---|---|---|---|---|
| Warden M, Y-Y Ho | 2025 | Infralimbic parvalbumin neural activity facilitates cued threat avoidance | https://osf.io/679jy/ | Open Science Framework, 679jy |

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
