## [Editor Report · eLife assessment]

This **important** study extends our understanding of how the medial prefrontal cortex regulates flexible action during adversity. The data provide **compelling** evidence of a role for prefrontal PV neuron activity in active avoidance. This builds on the general idea that these neurons play a role in flexible behavior and demonstrates this in the context of freezing/avoidance conflict. The overall findings contribute to our understanding of mechanisms that support aversively motivated instrumental learning and may provide insight into both stress vulnerability and resilience processes. This work will be of interest to those interested in learning, aversive motivation, interneuron and/or prefrontal cortex function, or conditions relates to these processes and mechanisms.

---

## [Referee Report · Reviewer #2 (Public Review)]

Summary:

This study examined the role of a prefrontal cortex cell type in active avoidance behavior. The authors conduct a series of behavioral experiments incorporating fiber photometry and optogenetic silencing. The results indicate that prefrontal parvalbumin (PV) neurons play a permissive role in performing signaled active avoidance learning, for which details are sorely lacking. Notably, infralimbic parvalbumin activity resolves incompatible defensive responses to threat by suppressing conditional freezing in order to permit active instrumental controlling responses. The overall findings provide a significant contribution to our understanding of mechanisms that support aversively motivated instrumental learning and may provide insight into both stress vulnerability and resilience processes.

Strengths:

The writing and presentation of data is clear. The authors use a number of temporally-relevant methods and analyses that identify a novel prefrontal mechanism in resolving the conflict between competing actions (freezing vs escape avoidance). The authors conduct an extensive number of experiments to demonstrate that the uncovered prefrontal mechanism is selective for the initiation of avoidance under threat circumstances, not reward settings or general features of movement.

Weaknesses:

The study exclusively focuses on parvalbumin cells, thus questions remain whether the present findings are specific to parvalbumin or applicable to other prefrontal interneuron subtypes. The exact mechanisms that coordinate infralimbic parvalbumin cell activity and threat avoidance behavior are not explored.

---

## [Referee Report · Reviewer #3 (Public Review)]

Summary:

Here the authors study the role of parvalbumin (PV) expressing neurons in the ventromedial prefrontal cortex (vMPFC) of mice in active avoidance behavior using fiber photometry and optogenetic inhibition.

Strengths:

The methods are appropriate, the experiments are well done, and the results are all consistent with the conceptual model in which vmPFC PV neurons inhibit freezing to enable avoidance movements. There are good controls to rule out a role for cue offset in triggering changes in PV neuron activity, or for a nonspecific role of vmPFC PV neurons in movement initiation.

Weaknesses:

Although potential mechanisms, i.e., the impact of PV neuron activity on the broader circuit, are discussed, they are not directly examined here. There is some discordance between changes in neural activity and behavior: in Figure 4C, the relationship between PV neuron activity and movement emerges almost immediately during learning, but successful active avoidance emerges much more gradually. Again, this is discussed and plausible explanations for this discrepancy are provided.

---

## [Author Response]

The following is the authors’ response to the original reviews.

Additional Discussion Points(1) There is not much exploration of potential mechanisms, i.e., the impact of PV neuron activity on the broader circuit. Additionally, the study exclusively focuses on PV cells and does not explore the role of other prefrontal populations, particularly those known to respond to cueevoked fear states. The discussion should consider how PV activity might impact the broader circuit and whether the present findings are specific to PV cells or applicable to other interneuron subtypes.

We have added an extensive discussion of potential mechanisms and the potential contributions of other interneuron subtypes:

“For example, PV neurons aid in improving visual discrimination through sharpening response selectivity in visual cortex (Lee et al., 2012). In prefrontal cortex, PV neurons are critical for task performance, particularly during performance of tasks that require flexible behavior such as rule shift learning (Cho et al., 2020) and reward extinction (Sparta et al., 2014). Further, PV neurons play an essential role in the generation of cortical gamma rhythms, which contribute to synchronization of selective populations of pyramidal neurons (Sohal et al., 2009; Cardin et al., 2009). Courtin et al (2014) showed that brief suppression of dorsomedial prefrontal (dmPFC) PV neural activity enhanced fear expression, one of the main functions of the dmPFC, by synchronizing the spiking activity of dmPFC pyramidal neurons (Courtin et al., 2014). This result is potentially relevant to our findings, but likely involves different circuit mechanisms because of the difference in timescale, targeted area, and downstream projection targets (Vertes, 2004). These and other studies support the idea that PV neural activity supports the execution of a behavior by shaping rather than suppressing cortical activity, potentially by selecting among conflicting behaviors by the synchronization of different pyramidal populations (Warden et al., 2012; Lee et al., 2014).

The roles of other inhibitory neural subtypes (such as somatostatin (SOM)-expressing and vasoactive intestinal peptide (VIP)-expressing IL GABA neurons) in avoidance behavior are currently unknown, but are likely important given the role of SOM neurons in gamma-band synchronization (Veit et al., 2017), and the role of VIP neurons in regulating PV and SOM neural activity (Cardin, 2018).”

(2) There is some discordance between changes in neural activity and behavior. For example, in Figure 4C, the relationship between PV neuron activity and movement emerges almost immediately during learning, but successful active avoidance emerges much more gradually. Why is this?

We have added extensive text to the discussion that addresses this issue:

“Interestingly, the rise in IL PV neural activity during movement does not require avoidance learning. IL PV neurons begin to respond during movement immediately after the animal has received a single shock in an environment, but learning to cross the chamber to avoid the signaled shock takes tens of trials. Why is there a discordance between the emergence of the IL PV signal during movement and avoidance learning?

The components underlying active avoidance have been debated over the years, but are thought to involve at least two essential behaviors – suppressing freezing, and moving to safety (LeDoux et al., 2017). Freezing is the default response of mice upon hearing a shock-predicting tone, and can be learned in a single trial (Ledoux, 1996; Fanselow, 2010; Zambetti et al., 2022). When a predator is in the distance, freezing can increase the chance of survival by reducing the chances of detection. However, a strategic avoidance behavior may prevent a future encounter with the predator altogether. The importance of IL PV neural activity in defensive behavior may be to suppress reactive defensive behaviors such as freezing in order to permit a flexible goaldirected response to threat.

The freezing suppression and avoidance movement components of the avoidance response are dissociable, both because freezing precedes avoidance learning, and because animals intermittently move prior to avoidance learning. Our finding that the rise in PV activity during movement emerges immediately after receiving a single shock, tens of trials before animals have learned the avoidance behavior, suggests that the IL PV signal is associated with the suppression of freezing. Further, IL PV neurons do not respond during movement toward cued rewards because in reward-based tasks there is no freezing response in conflict with reward approach behavior.”

(3) vmPFC was defined here as including the infralimbic (IL) and dorsal peduncular (DP) regions. While the role of IL has been frequently characterized for motivated behavior, relatively few studies have examined DP. Perhaps the authors are just being cautious, given the challenges involved in the viral targeting of the IL region without leakage to nearby regions such as DP. But since the optical fibers were positioned above the IL region, it is possible that DP did not contribute much to either the fiber photometry signals or the effects of the optogenetic manipulations. Perhaps DP should be completely omitted, which is more consistent with the definitions of vmPFC in the field.

Yes, we included DP to be cautious as our viral expression sometimes leaks into DP, though the optic fiber targets IL. We have replaced vmPFC with IL throughout the manuscript.

(4) In the Discussion, the authors should consider why PV cells exhibit increased activity during both movement initiation and successful chamber crossing during avoidance. While the functional contribution of the PV signal during movement initiation was tested with optogenetic inhibition, some discussion on the possible role of the additional PV signal during chamber crossing is of interest readers who are intrigued by the signaling of two events. Is the chamber crossing signal related to successful avoidance or learned safety (e.g., see Sangha, Diehl, Bergstrom, Drew 2020)?

IL PV neural activity starts to increase at movement initiation, peaks at chamber crossing (when movement speed is highest), and decreases after chamber crossing (Figure 1E). Thus, the increase in PV neural activity at movement initiation and at chamber crossing are different phases of the same event.

We think this signal is unlikely to be a safety signal, and have added text to the discussion to clarify this issue:

“We think the IL PV signal is unlikely to be a safety signal (Sangha et al., 2020). First, the PV signal rises during movement not only in the avoidance context, but during any movement in a “threatening” context (i.e. a context where the animal has been shocked). For example, PV neural activity rises during movement during the intertrial interval in the avoidance task. Further, the emergence of the PV signal during movement happens quickly – after the first shock – and significantly before the animal has learned to move to the safe zone. This suggests a close association with enabling movement in a threatening environment, when animals must suppress a freezing response in order to move. Additionally, the rise in PV activity was specifically associated with movement and not with tone offset, the indicator of safety in this task. Finally, if IL PV neural activity reflects safety signals one would expect the response to be enhanced by learning, but the amplitude of the IL PV response was unaffected by learning after the first shock.”

(5) The primary conclusion here that PV cells control the fear response should be considered within the context of prior findings by the Herry laboratory. Courtin et al (2014) demonstrated a select role of prefrontal PV cells in the regulation of fear states, accomplished through their control over prefrontal output to the basolateral amygdala. The observations in this paper, which used both ChR2 and Arch-T to address the impact of vmPFC PV activity on reactive behavior, are highly relevant to issues raised both in the Introduction and Discussion.

Courtin et al (2014)’s finding is very important. We did not discuss this paper originally because Courtin et al. is about dmPFC, which has a different role in fear processing than IL/vmPFC. We have added text about this finding to the discussion:

“Courtin et al (2014) showed that brief suppression of dorsomedial prefrontal (dmPFC) PV neural activity enhanced fear expression, one of the main functions of the dmPFC, by synchronizing the spiking activity of dmPFC pyramidal neurons (Courtin et al., 2014). This result is potentially relevant to our findings, but likely involves different circuit mechanisms because of the difference in timescale, targeted area, and downstream projection targets (Vertes, 2004).

Additional analyses(1) As avoidance trials progress (particularly on days 2 and 3), do PFC PV responses attenuate? That is, does continued unreinforced tone presentations lead to reduced reliance of PV cellmediated suppression in order for successful avoidance to occur?

We added Figure 1—Figure supplement 1M and 1N and a sentence on page 5: “IL PV neural activity during the avoidance movement was not attenuated by learning or repeated reinforcement (Figure 1—Figure supplement 1M and N, N = 8 mice, p = 0.8886, 1-way ANOVA).” We only included data from days 1 and 2, since we started to introduce short and long tone trials on day 3 which might interfere.

(2) In Figure 3D, it would be very informative and further support the claim of "no role for movement during reward" if the response of these cells during the "initiation of movement during reward-approach" was shown (similar to Figure 1F for threat avoidance).

Thank you for the question. We added Figure 3—Figure supplement 1B and C to show IL PV neural activity aligned to initiation of movement during reward-approach. IL PV activity decreased after movement initiation for reward approach (N = 6 mice, p=0.0382, paired t-test). This further solidifies our claim that IL PV neuron activity only increases for threat avoidance.

**Reviewer 1 (Recommendations For The Authors):**
(1) Fig1G shows the average response of PV cells during chamber crossing on an animal-toanimal basis. It would be informative to also see a similar plot for movement initiation.

We have added the suggested figure in Figure 1—Figure supplement 1B.

(2) In the Results section (Page 5), there is a small issue with the logic. It says: "As vmPFC inactivation impairs avoidance behavior, the activity of inhibitory vmPFC PV neurons might be predicted to be low during successful avoidance trials." As opposed to "low", it should say "high", right? If inhibition impairs avoidance, then high responding by these cells would be presumed to drive the avoidance response, as supported by your findings.

We have re-worded the text in this section. Based on prior findings that IL inactivation impairs avoidance (Moscarello et al., 2013), we predicted that inhibitory PV neurons would be less active during avoidance, because activating these neurons could suppress IL. However, we found that they were selectively active during avoidance.

(3) In the caption/legend for Fig1E, it says that the "black ticks" indicate "tone onset". But it should say "movement initiation".

We thank the reviewer for pointing out this error. The ticks do indicate tone onset, and we have corrected the figure to reflect this.

**Reviewer 2 (Recommendations For The Authors):**
(4) Perhaps replace the term 'good outcomes' with 'reinforcing outcomes' or simply 'reinforcement'.

Thank you for the suggestion. We have replaced ‘good outcomes’ with ‘reinforcing outcomes’.

**Reviewer 3 (Recommendations For The Authors):**
(5) It would be useful to provide some (perhaps speculative) explanation for the discordance between the PV activity-movement relationship and success of active avoidance in Fig. 4C

We have added text to the discussion that addresses this issue:

“Interestingly, the rise in IL PV neural activity during movement does not require avoidance learning. IL PV neurons begin to respond during movement immediately after the animal has received a single shock in an environment, but learning to cross the chamber to avoid the signaled shock takes tens of trials. Why is there a discordance between the emergence of the IL PV signal during movement and avoidance learning?

The components underlying active avoidance have been debated over the years, but are thought to involve at least two essential behaviors – suppressing freezing, and moving to safety (LeDoux et al., 2017). Freezing is the default response of mice upon hearing a shock-predicting tone, and can be learned in a single trial (Ledoux, 1996; Fanselow, 2010; Zambetti et al., 2022). When a predator is in the distance, freezing can increase the chance of survival by reducing the chances of detection. However, a strategic avoidance behavior may prevent a future encounter with the predator altogether. The importance of IL PV neural activity in defensive behavior may be to suppress reactive defensive behaviors such as freezing in order to permit a flexible goaldirected response to threat.

The freezing suppression and avoidance movement components of the avoidance response are dissociable, both because freezing precedes avoidance learning, and because animals intermittently move prior to avoidance learning. Our finding that the rise in PV activity during movement emerges immediately after receiving a single shock, tens of trials before animals have learned the avoidance behavior, suggests that the IL PV signal is associated with the suppression of freezing. Further, IL PV neurons do not respond during movement toward cued rewards because in reward-based tasks there is no freezing response in conflict with reward approach behavior.”

(6) I don't really understand what is shown in Figure 4D -- exactly what time points does this represent? Was habituation performed everyday?

Figure 4D shows data from the approach task, not the avoidance task. This data is from welltrained mice, not the first day of training on this task. There was a pre-task recording period every day.

(7) Why was optogenetic inhibition only delivered from 0.5-2.5 sec after the tone cue?

We wanted to avoid any possibility that perception of the tone would be disrupted, so we delayed the onset of optogenetic inhibition. We chose 0.5 sec onset because animals typically begin to move ~1 second after tone onset.

(8) The regression analysis with shuffled time points is not well explained -- some additional methodological details are needed (Fig. 2H).

We added the following to the methods section to provide a clearer explanation:

“DF/F (t) was modeled as the linear combination of all event kernels. Given the event occurrence time points of all event types, we can use linear regression to decompose characteristic kernels for each event type. Kernel coefficients of the model were solved by minimizing the mean square errors between the model and the actual recorded signals. To prove that kernel ki is an essential component for the raw calcium dynamics, we compared the explanation power of the full model to the reduced model where the time points of the occurrence of event ki were randomly assigned. Thus, the kernel coefficients should not reflect the response to the event in the reduced model.

Editor's notes:- Should you choose to revise your manuscript, please include full statistical reporting including exact p-values wherever possible alongside the summary statistics (test statistic and df) and 95% confidence intervals. These should be reported for all key questions and not only when the pvalue is less than 0.05.

Thank you for pointing this out. We have included all the test statistics and exact p values as suggested.

- Please note the sex of the mice and distribution of sexes in each group for each experiment.

We have added the sex of mice for all experiments in the methods section.